# Is Prolonging the Lifetime of Passive Durable Products a Low-Hanging Fruit of a Circular Economy? A Multiple Case Study

**Mohamad Kaddoura** [1,2,*], **Marianna Lena Kambanou** [3], **Anne-Marie Tillman** [1] and **Tomohiko Sakao** [3]

[1] Division of Environmental System Analysis, Department of Technology Management and Economics, Chalmers University of Technology, SE-412 96 Gothenburg, Sweden
[2] CIRAIG, Department of Chemical Engineering, École Polytechnique de Montréal, P.O. Box 6079, Succ, Centre-ville, Montréal, QC H3C 3A7, Canada
[3] Division of Environmental Technology Management, Department of Management and Engineering, Linköping University, SE-581 83 Linköping, Sweden
* Correspondence: mohamad.kaddoura@polymtl.ca; Tel.: +1-438-926-1711

**Abstract:** Extending the lifetime of passive products, i.e., products that do not consume materials or energy during the use phase, by implementing product-service systems (PSS) has a potential to reduce the environmental impact while being an attractive and straightforward measure for companies to implement. This research assesses the viability of introducing PSS for passive products, by documenting five real product cases of prolonging the lifetime through repair or refurbishment and by quantifying, through life cycle assessment (LCA) and life cycle costing (LCC), the change in environmental and economic outcome. The environmental impact (measured as global warming potential over the life cycle) was reduced for all cases because extraction and production dominated the impact. This reduction was 45–72% for most cases and mainly influenced by the number of reuses and the relative environmental burden of the components whose lifetime was prolonged. The costs for the company (measured as LCC from the manufacturer's perspective) decreased too by 8–37%. The main reason that costs reduced less than the environmental impact is that some costs have no equivalent in LCA, e.g., administration and labor costs for services. The decreases in both LCA and LCC results, as well as the willingness of the companies to implement the changes, demonstrate that this measure can be financially attractive for companies to implement and effectively contribute to a circular economy.

**Keywords:** life cycle assessment; life cycle costing; resource efficiency; circular economy; product-service system; refurbishment; repair

## 1. Introduction

The unsustainable use of resources, driven by population growth and industrial and economic expansion, could triple by the year 2050, compared to the levels at the turn of the 21st century [1]. To ensure prosperity, there is a clear need to decouple resource use from economic growth. A circular economy (CE) is a vision for such a decoupling. In a circular economy, resources are restored or regenerated, energy is renewable, and waste and toxic chemicals are reduced through the design of materials, products, systems, and business models [2,3]. This is an umbrella concept that encompasses a variety of strategies, and one of the key strategies is to extend, or prolong, the lifetime (i.e., usable life) of products or their components [4,5].

One way to operationalize the extension of a product's lifetime is through the addition or integration of a service and the subsequent transformation of a company's offering from a product into a product service-system (PSS) [6–8]. PSS is defined as "an integrated bundle of products and services which aims at creating customer utility and generating value" [9] and includes (i) product-oriented services where services are added to a product, (ii) use-oriented services where the provider maintains ownership and the product's use is sold, and (iii) result-oriented services where the result of the product is sold [10]. Although the idea is simple, it is noted in literature that not every PSS contributes to a CE, e.g., [6], and for this to happen, two key conditions have to be in place. The first is that a PSS, or other circular measure, must be an attractive business for companies so that it actually gets implemented [11–13], and the second is that it must contribute at least to relative resource decoupling [6].

There is a large amount of academic literature on CE barriers (e.g., [11,14,15]) concerning the first point. A common major finding in this literature is that companies lack willingness, awareness, and resources to engage with CE, and that they often perceive CE strategies as too risky and lacking economic viability [11,14,15]. Furthermore, on a global level, CE is limited by the structure of the existing formal economy and the norms and values of a society as well as the institutional conditions that influence the decision making of all actors [16,17]. Therefore, policies at all governmental levels [11,14,15] and actions across globalized supply chains and institutions are needed to support a transition to CE [16]. However, in the absence or in wait of such policies and actions, and to overcome the aforementioned barriers, companies need solutions that are economically viable and can be implemented in a short-time perspective without the input of major resources or restructuring of the existing supply chain. We propose that prolonging the lifetime of passive durable products through PSS could be such a solution because it is not as complex as other CE strategies, e.g., the creation of sharing platforms or constant administration of active product or completely redesigning the offering as advocated by result-oriented services. Furthermore, although the potential of a PSS to contribute to resource reduction is dubious both per unit and due to rebound effects [6], the extension of the lifetime of passive durable products, i.e., products that do not consume energy or materials during the use phase, has been found by researchers to have high potential [18–20]. The reason is that the majority of the environmental impacts occur during extraction of raw materials and production, and consequently, extending the lifetime of the product makes environmental sense (ibid.). For example, Mont et al. [21] identified baby prams as a relevant product group for applying a PSS leasing model. Heiskanen and Jalas [22] also reported on increased eco-efficiency of renting skiing equipment and Gutowski et al. [23] found environmental benefits based on life cycle assessment (LCA) on extending the lifetime of office furniture. However, only one article was found to quantify both the economic and environmental outcomes of prolonging the lifetime of passive products through PSS; Lindahl et al. [24] demonstrated substantial reduction of both LCA and LCC through reuse of the core plugs for paper rolls at paper mills.

Therefore, the purpose of this article is to quantitatively explore the potential of extending the lifetime of passive durable products through PSS through five real cases at three case companies. Lifetime extension of passive durable products through PSS is in this work seen as the low-hanging fruit of a CE, as described by theory further elaborated on in Section 2. The present study contributes new knowledge to the fields of circular economy and PSS through:

- documenting cases where the lifetime of durable passive products has been prolonged through the addition of services;
- quantitatively assessing the implications of the measures from an environmental perspective using life cycle assessment (LCA), and from a cost perspective using life cycle costing (LCC), and;
- discussing the potential of prolonging the lifetime of passive durable products to contribute to a more circular economy while being economically viable for the manufacturer.

## 2. Background and Motivation

Although prolonging the lifetime of products, or their components, is an important strategy to achieve a more resource-efficient society [25], lifetime extension is only valid when the avoided impacts from production, e.g., energy use and the subsequent reduction in impact upstream such as raw material extraction and transportation, are lower than the impacts from producing more durable components or treating the returned product to extend its lifetime [20,26]. For many products, lifetime extension reduces environmental impacts per unit of function, but there are exceptions. For example, there can be environmental benefits to replacing active products, which consume energy and consumables during the use phase, with more energy-efficient ones rather than prolonging their lifetime [19,27]. Case-by-case decisions are necessary [28] for these types of products. Similarly, replacing single-use products, e.g., diapers and incontinence products, with more durable ones has also demonstrated both positive and negative results [18]. Passive durable products, however, hold a larger and more consistent potential for environmental improvement because their environmental impact is dominated by the production phase, usually the production of materials, while their physical lifetime is not fully exploited [20]. By prolonging the lifetime of such products, more function is delivered from the same product; therefore, the impact per function delivered decreases, which leads to relative resource decoupling. This is in line with the conclusions from a number of studies concerning passive durable products, which found environmental improvements when extending product lifetime (e.g., [24,29–33]).

This is important because it essentially means that companies that have passive durable products can opt for strategies and business models that extend the lifetime of their products with high confidence that they will be contributing to a more circular economy and resource efficiency per unit of function delivered. If clearly communicated to companies, this condition will make the process of choosing a circular strategy, or measure, less complex, and thus reduces one of the key barriers to implementing circular offerings [11,14].

For companies, however, decreasing the environmental impacts associated with their products is only one part of their motivation; the offerings also need to be financially viable [34]. PSS, which advocates the addition or integration of services with a product, could support the economic viability of lifetime prolongation. Although one of the initial foci in PSS literature has been its potential to bring about joint environmental and economic benefits through the integration of products and services, in its evolution, the focus lies mainly on the attributes of PSS as a competitive business strategy [7]. PSS has gained traction with companies because it can give them a competitive advantage, better customer knowledge, and new sources of income [35–37]. Therefore, achieving lifetime extension of passive durable products through PSS, which has great potential to contribute to environmental impact reduction and can be a profitable proposition, theoretically holds water. However, redesigning company offerings toward PSS in not without challenges [38,39], and the higher the level of integration of the product and the service, e.g., result-oriented versus product-oriented, the more change within the company is required to do it successfully [40]. Therefore, focusing on product- or use-oriented services for the implementation of CE strategies, as less complex than result-oriented services, may lead to a higher adoption rate.

A lot of research in the circular economy field is either qualitative or focuses on the environmental perspective [13]. Quantitative studies, and especially research that combines the environmental and economic perspectives of extending the lifetime of products, are few in the CE context [19] and in the PSS context [24]. Further, there is a lack of assessment studies of real industrial cases, and most of what can be found in literature is based on theoretical cases [41]. Therefore, there is a need to go beyond theory and quantitatively assess the benefits, or the lack of them, in real case studies, both from the perspective of the environment and from the perspective of the companies that are actually bringing offerings to the market.

The tools most often used for joint assessment are LCA and LCC, e.g., [24]. Although there are numerous assessment studies that combine LCA and LCC, as reviewed by Miah et al. [42], the majority

of those studies are from the building, energy, and waste and water sectors. The ones that focus on prolonging the lifetime of products using both LCA and LCC analyze the effect of increasing the durability of active products instead of replacing them with more efficient products (e.g., [19,43,44]). Therefore, the contribution of the present study, which uses LCA to assess environmental impacts and avoid burden shifting between life cycle stages, and LCC for the manufacturer to understand the economic consequences of extending the lifetime of passive durable products, is necessary.

## 3. Materials and Methods

### 3.1. Overview

The research is based on five different cases of passive durable products from three Swedish companies (Companies A, B, and C). Companies A and B are small to medium-sized. Multiple cases were used because they give a stronger base for theory building as well as context for research [45]. The criterion for choosing the companies was that they brought passive products to the market through traditional product sales models but were willing to explore the option of transforming their offering into PSS. The five products were redesigned into more circular offerings that included services, as summarized in Table 1. Although all products were passive, they varied with respect to complexity (number of components and materials) and lifetime. In order to prolong product lifetime, all offerings required changes to products and, in some cases, also to the business model. To understand the consequences of extending product lifetime, LCA and LCC for the manufacturer of the more circular offering and the existing business-as-usual (BAU) product were compared. The selection of the products and the process of choosing circular measures for the products were done in dialogue between the companies and the researchers. The products were selected because they were important to the company's portfolio, they could be easily disassembled, and they required minor repair/refurbishment to restore them to a good condition. The researchers also collaborated with the companies when obtaining data for LCA and LCC analyses as well as in the discussion of the results. These interactions cannot be broken down into measurable events, as they included meetings, emails, and telephone calls, but the interactions contributed to analyzing the results and to understanding the economic viability of the products beyond life cycle costs as well as some of the potential rebound effects of the more circular offerings.

**Table 1.** Overview of the business-as-usual (BAU) and circular offerings and business models of the products analyzed in the study.

| Product | BAU Offering | BAU Business Model | Circular Offering | Circular Business Model |
|---|---|---|---|---|
| Beach flag | 410 cm tall flag used once | Product sales | 410 cm tall flag used 10 times | Product lease |
| Event tent | 300 × 345 × 300 cm tent used once | Product sales | 300 × 345 × 300 cm tent used 10 times | Product lease |
| Recycling bin | 2 × 85 L bins surviving 7.5 years | Product sales | 2 × 85 L bins surviving 15 years | Product sales + repair contract |
| Locker | 300 × 300 × 47 cm locker used for 10 years | Product sales | 300 × 300 × 47 cm locker used for 20 years | Product sales + refurbish contract |
| Waste inlet | Waste inlet used for 30 years | Product sales + maintenance and repair contract | Waste inlet with modified door used for 30 years | Product sales + maintenance and repair contract |

For all circular offerings, product lifetime was extended through repair or refurbishment. Repair is defined as restoring the product "so it can be used in its original function," while refurbishment is defined as "restoring an old product and bringing it up to date" [46]. In one case (Company C), a modular design that allowed for more efficient repair was implemented. The types of PSSs used as means to achieve the circular offerings were product-oriented services and use-oriented services

(Table 1). Detailed life cycle inventory data for the products are presented in the Supplementary Material. The LCC data, however, are not disclosed for confidentiality reasons.

## 3.2. LCC and LCA

**LCC:** LCC was performed from a manufacturer's perspective (LCC$_{man}$), according to the IEC 60300-3-3:2017 guide [47]. Cost data were provided by the companies, and where historic data were not available, they were substituted with expert estimations. The costs were aggregated into stages to allow for a more integrated assessment of results later. The subcategories and the cost elements of each are illustrated in Table 2. Since the lifetime of the products was fairly short, no discounting was done for costs that occur in the future (e.g., maintenance, repair, and end-of-life).

**Table 2.** Life cycle phases used in the LCC$_{man}$ and life cycle assessment (LCA) analyses and the included activities. Activities in **bold** type are common between LCC$_{man}$ and LCA.

| Life Cycle Phase | LCC$_{man}$ | LCA |
|---|---|---|
| **Design and development** | -Research and development<br>-Graphic design | |
| **Marketing and sales** | -Marketing<br>-Order reception<br>-Transport<br>-Sales | |
| **Production** | -Administration<br>-Product reception<br>**-Product [1]**<br>**-Transport (upstream) [2]**<br>-Storage | **-Raw material extraction and production**<br>**-Manufacturing activities**<br>**-Transport (upstream) [2]** |
| **Distribution** | -Assembling<br>-Installation<br>**-Transport (downstream)**<br>-Administration | **-Transport (downstream)** |
| **Use [3]** | **-Maintenance** | **-Maintenance** |
| **Repair/refurbish** | -Administration<br>-Inspection<br>**-Product [1] (cost for spare part)**<br>**-Repair/refurbish activities**<br>**-Transport [4]** | **-Raw material extraction and production (spare part)**<br>**-Manufacturing activities (spare part)**<br>**-Repair/refurbish activities**<br>**-Transport [4]** |
| **End-of-life** | **-Transport (waste collection)**<br>**-Incineration** | **-Transport (waste collection)**<br>**-Incineration** |

[1] Product cost includes the cost for raw material, the production cost, and all upstream transports before arrival to the first-tier supplier. [2] Transport (upstream) refers to the transport from the first-tier supplier to the manufacturer. Other transports from higher-tier suppliers are included in the product cost (for LCC) and the corresponding raw material extraction and production activities (for LCA). [3] Since the products in the present study were passive, the operation had neither a cost nor an environmental burden for, e.g., use of energy or auxiliary materials. However, one of the companies occasionally incurs a maintenance cost for personnel travel to a site to survey and assess potential maintenance needs. For this reason, maintenance was used as the category instead of use. This category includes the transportation and labor cost in the LCC, and emissions from transportation to the site in LCA. [4] This includes the collection transport from the customer, the redistribution transport back to the customer, as well as the transport of the spare part from the first-tier supplier.

**LCA:** LCA was done according to the ISO 14040:2006 standard [48] and modeled using the OpenLCA software [49]. Bills for materials and data on logistics were provided by the companies. For upstream activities, the data were sourced from the Ecoinvent 3.3 database [50], and for textile components from Roos et al. [51].

**System boundaries:** Recycling at the end-of-life (EOL) stage was modeled using the cut-off method for both LCA and LCC [52], which means that the environmental impact from collection and recycling was set outside the system boundaries, and no credit was given for recycled materials at the EOL. In line with this, the use of recycled material as an input was not burdened with any environmental impact other than that of the recycling activities and subsequent transportation. The burden from

transportation and emissions during incineration was included for incineration with energy recovery, but no credit was given for energy recovery as avoided emissions. The cut-off method was used to align LCA and LCC. For instance, the LCC used in the present study includes the fee paid for incinerating products but not the revenues from selling the delivered energy.

**Impact assessment methods:** Three midpoint categories from the CML impact assessment package (v4.4, January 2015) [53] were selected to assess the impact of emissions, namely, (i) global warming potential (GWP) measured in kg $CO_2$-eq, (ii) acidification potential (AP) measured in kg $SO_2$-eq, and (iii) eutrophication potential (EP) measured in kg $PO_4^{3-}$-eq. These indicators, however, are often cross-correlated [54]; therefore, conclusions related to the dominant life cycle phases in LCA were drawn based on the GWP indicator. No impact assessment method was used to aggregate the use of different resources due to the instability of indicators for resource use in LCA [55]. The use of the raw materials iron ore and bauxite was used instead to indicate material resource use, depending on whether steel or aluminum was the main material in the product. The same procedure was followed for the land-use indicators, especially for bio-based materials, where land use might be in conflict with the results of the GWP. Accordingly, intensive forest occupation measured in $m^2$. year was used as a land-use indicator when wood products were under study.

**Sensitivity analysis:** The number of re-uses of the products in the original study was estimated by the suppliers, and this number was assumed to be an accurate representation of what could be, or already is, implemented. However, no standardized method currently exists for tracking the number of remanufacturing cycles a given product has undergone. Therefore, a sensitivity analysis of the durable components' reuse times was conducted.

**LCC and LCA comparison:** To present the results from both tools in a comparable way, cost and environmental burdens were aggregated according to life cycle phases, as shown in Table 2. Activities in bold type are those accounted for in both the LCC$_{man}$ and LCA. The table was constructed by adjusting the common elements of LCC and LCA as presented by Rebitzer [56] to our specific cases. *Repair/refurbish* is an additional life cycle phase when dealing with circular offerings, and this phase includes the reverse logistics for the used product, the production of spare parts, and the distribution of the repaired/refurbished product. It should also be noted that the costs related to *design and development* and *marketing and sales* were mainly labor costs, which have no equivalent environmental burdens in LCA. The notion is that labor incurs no extra environmental burden since people cause the same environmental impact, irrespective of whether they work or not (e.g., for housing, eating and other activities).

The production phase has also been divided into three different parts to allow for an advanced analysis. "Production D" refers to the production of durable components, i.e., the parts whose lifetime is extended in the circular offering. "Production R" refers to the original production of the component being replaced in the circular offering. Finally, "Production S" in the repair/refurbish phase refers to the production of spare parts. The sub-categories included in those productions are according to those in "Production" in Table 2, e.g., this includes some labor costs in the LCC$_{man}$ that are not included in the LCA. Table 3 shows what is included in each production part for the analyzed products.

**Table 3.** Overview of the three production parts for the products analyzed in the study.

| Product | Production D (BAU) | Production D (Circular) | Production R (BAU) | Production R (Circular) | Production S (Circular) |
|---|---|---|---|---|---|
| Beach flag | 10 pole and weights | 1 pole and weights | 10 textiles | 1 textile | 9 textiles |
| Event tent | 10 frame and weights | 1 frame and weights | 10 textiles | 1 textile | 9 textiles |
| Recycling bin | 2 bodies | 1 body | 2 lids | 1 lid | 2 lids |
| locker | 2 Bodies (excluding the paint) | 1 Body (excluding the paint) | 2 paints | 1 paint | 1 paint |
| Waste inlet | 1 waste inlet (excluding the door) | 1 waste inlet (excluding the door) | 3 doors | 1 door | 2 doors |

## 4. Case Studies

### 4.1. Refurbishment of Beach Flag and Event Tent

#### 4.1.1. Case Product and Company

Company A is a supplier of physical and digital visibility solutions for indoor and outdoor use that are used by other companies to provide brand exposure. Company A is responsible for the marketing, sales, and graphic design. It sources parts from various suppliers, assembles the order, and is responsible for arranging transportation to the customers, installation, and storage between uses. Two products were selected, an event tent and a beach flag (Figure 1), which are often used for one event only before EOL.

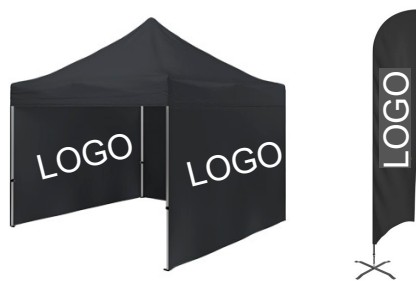

**Figure 1.** Event tent (**left**) and beach flag (**right**).

The event tent consists of a $300 \times 345 \times 300$ cm aluminum frame covered on three sides and on the top with textile, and it is supported with steel weights. The beach flag is a 410 cm tall flag with a pole made of a fiberglass composite, a steel cross-base, and a textile of $72 \times 360$ cm attached to the pole. In the BAU offering, a flag is sold to the user who disposes of it after its first use. In the circular offering, the ownership of the flag is, instead, retained by Company A, which leases it to the customer for one event and collects it. Company A then refurbishes the product by reusing the pole and cross-base and replacing the textile, which is customized for each event. The refurbishing process also includes cleaning the pole with water and ensuring that it is in a good condition. Each pole is expected to maintain a good condition for 10 events. The concept is similar for the event tent; the durable aluminum frame and steel weights are reused, while the textile is changed during refurbishment. The tent frame and weights are also estimated to be used 10 times before EOL.

The manufacturing of the metal components takes place in China, from where they are transported to a warehouse in Sweden. At the warehouse, the components are equipped with textile parts, also from China, and distributed to customers. The refurbishment of the products takes place in Sweden. At the EOL, the metal parts are assumed to be sent to recycling, while the textile and fiberglass parts are sent to incineration with energy recovery.

The functional unit for both the LCC and the LCA was set to one item used for one event.

#### 4.1.2. Results

Figure 2a,b shows the overall results of the $LCC_{man}$ and LCA for the beach flag and event tent, respectively, normalized to the BAU offering. Figure 2c–e shows the contribution from the different life cycle phases to $LCC_{man}$ and GWP. The $LCC_{man}$ was reduced by 12–18% in the circular offering. The environmental impact was reduced much more, by 45–88%, with the largest reduction for the indicator "amount of iron ore" used over the life cycle. The main reasons for the difference in reduction between $LCC_{man}$ and LCA are due to: (i) costs not related to production, and (ii) costs and environmental impacts of the dominant phase.

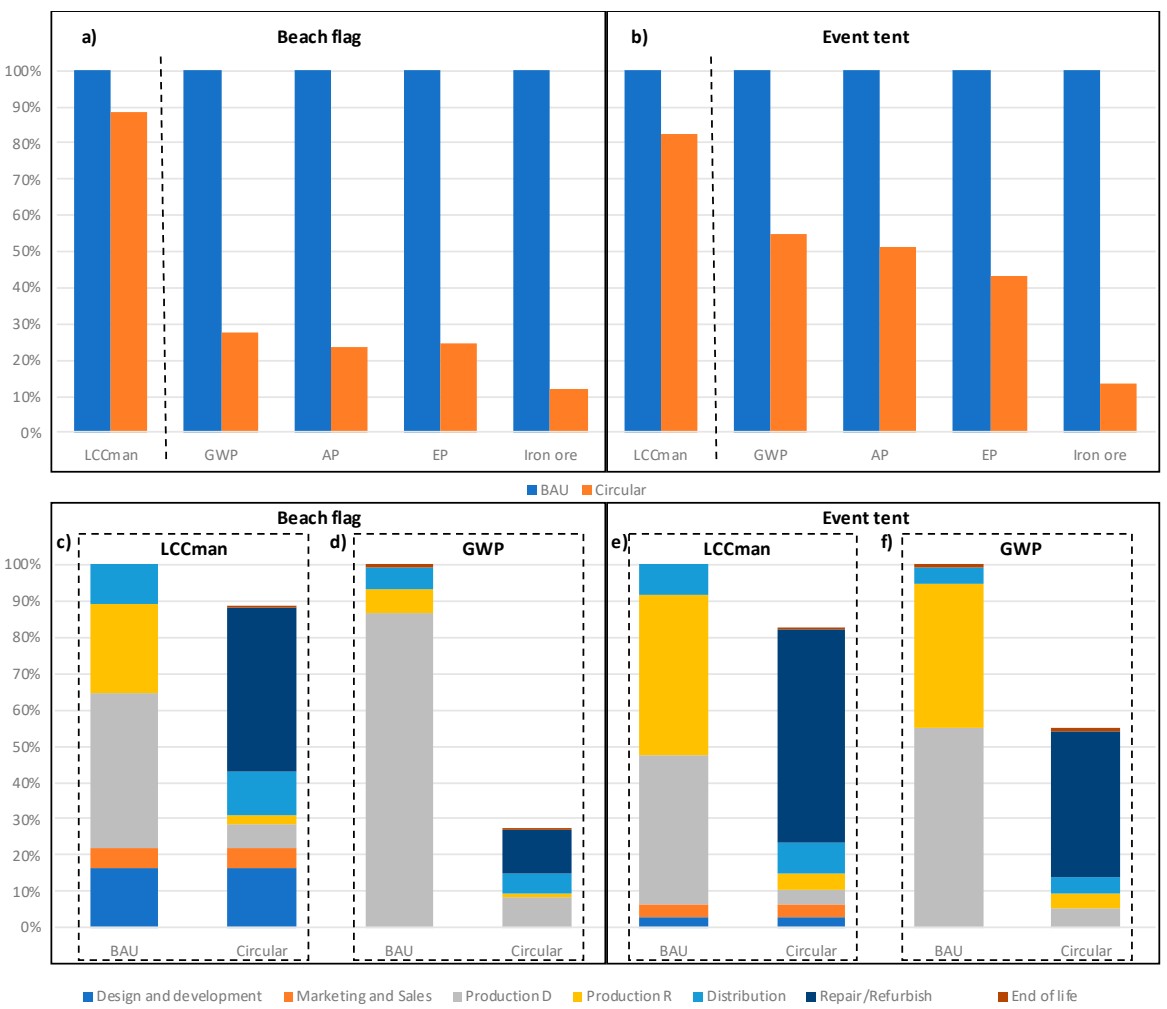

**Figure 2.** LCC$_{man}$ and LCA results for the BAU and circular offerings of the (**a**) beach flag and (**b**) events tent. The contribution of each life cycle phase (in accordance with the categorization in Table 2) towards (**c**) LCC$_{man}$ and (**d**) global warming potential (GWP) for the beach flag and (**e**) LCC$_{man}$ and (**f**) GWP for the events tent is shown. LCC$_{man}$ is in SEK, GWP is in kg CO$_2$-eq, AP is in kg SO$_2$-eq, EP is in kg PO$_4^{3-}$-eq, and iron ore is in kg iron ore. Production D denotes cradle-to-gate production of durable parts, Production R is cradle-to-gate production of original textile component, and repair/refurbish includes the cradle-to-gate production of textile spare parts. All charts are normalized with respect to the BAU offering.

**(i) Costs not related to production:** The first observation is that some costs, mainly labor-related ones, such as marketing and sales, design and development, and distribution costs, were the same for the BAU and circular offerings, while others changed (Figure 2c,e). The reason is that the former were independent of the production process and its raw materials. This limits the cost-saving potential when implementing a circular offering. The environmental impacts, however, are more tied to the production process and materials.

**(ii) Costs and environmental impacts of the dominant phase:** The cost in the BAU offering was dominated by the price of the physical components (Figure 2c,e). Together, the cost of the durable parts (Production D) and the textile parts (Production R) amounted to 66% and 84% of the total life cycle cost for the BAU flag and tent, respectively. Cost savings in the circular offering were limited since the price of the textile components replaced during refurbishment is high (Production S). This is in contrast to the environmental profile (Figure 2d,f). In that profile, the GWP for both products was dominated by Production D of the durable parts (fiberglass pole and steel cross-base for the flag and aluminum frame

and steel frame weights for the tent) in the BAU offering (87% and 74%, respectively). This is the reason why the environmental improvement in the circular offering is greater. Nevertheless, the production of textile spare parts (Production S) was not negligible in terms of GWP in the circular offering.

Finally, a comparison of the results for the event tent and the beach flag shows that reusing the durable components of the tent yields higher cost savings than reusing the durable components of the beach flag. This observation exemplifies something that could also be logically hypothesized. For larger products, the share of production-related costs of the component whose lifetime is prolonged becomes more dominant than the labor-related costs, which results in greater potential for savings. On the other hand, from an environmental perspective, the component being replaced when the tent is refurbished (textile) carries a larger share of the GWP, compared to the flag, which is why the relative environmental gains were greater for the flag.

### 4.1.3. Sensitivity Analysis

In the present study, it was assumed that the durable components of the beach flag and event tent were reused 9 times during product lifetime, and for that reason, this parameter was varied in the sensitivity analysis.

Figure 3 shows the results of the sensitivity analysis for Company A. The x-axis shows the number of uses (one use is the original BAU and 10 uses are the original circular offering), and the y-axis shows the normalized GWP and $LCC_{man}$. The figure shows that the GWP and $LCC_{man}$ decrease as the number of uses increase. The decrease in the GWP for both the beach flag (Figure 3a) and the event tent (Figure 3c) drops sharply at the beginning, then flattens towards the end. Most of the environmental benefits of the circular offering were obtained after 4–5 uses. However, the improvements became less prominent if the product was reused 5 or 10 times. The $LCC_{man}$ for the beach flag (Figure 3b) and the event tent (Figure 3d) decreased in a more linear manner, but the rate of decrease was limited. This is explained by the fixed labor costs, i.e., design and development and marketing and sales costs, limiting the decrease, and the high cost of the textiles (for the spare parts).

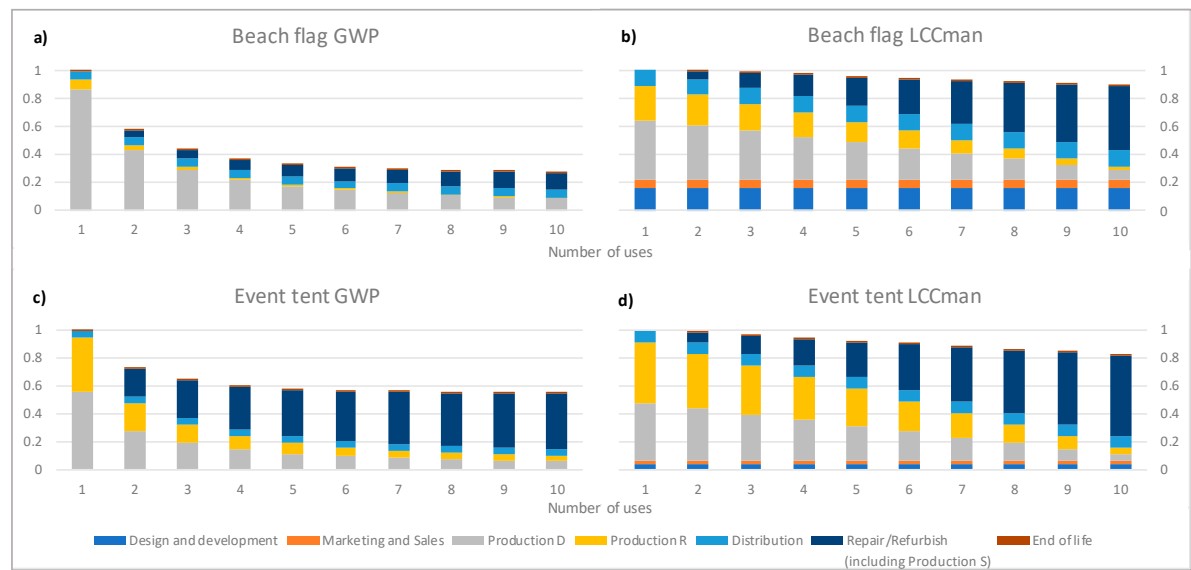

**Figure 3.** The sensitivity analysis results for the varied number of uses. (**a**) shows the variation of GWP, and (**b**) shows the variation of $LCC_{man}$ for the beach flag as the number of uses varied. (**c**) shows the variation of GWP, and (**d**) shows the variation of $LCC_{man}$ for the events tent as the number of uses varied. GWP is in kg $CO_2$-eq, and $LCC_{man}$ is in SEK. All charts are normalized with respect to the single use case.

### 4.2. Repair of Recycle Bin and Locker

#### 4.2.1. Case PRODUCT and Company

Company B is a supplier of storage furniture for public spaces. Company B designs the products and is responsible for marketing and administration. Manufacturing, distribution, and storage are subcontracted to suppliers, and retailers sell the products. The selected products were a recycling bin and an office storage locker (Figure 4). The bin consisted of a steel body and a set of lids made from medium-density fiberboard (MDF). The current estimated average lifetime of the bin is 7.5 years, after which it is disposed of mainly because the lid becomes battered. The circular offering of the recycling bin consisted of a repair option that could be purchased, in which the lid would be exchanged. This was estimated to prolong the lifetime of the bin to 15 years, as opposed to 7.5 years in the BAU offering. The lid was estimated to be changed twice during the 15-year lifetime.

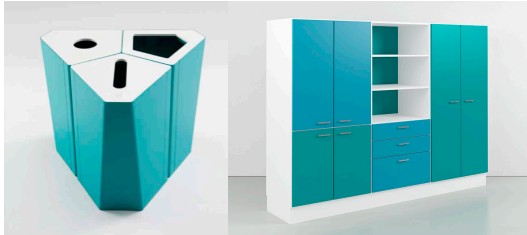

**Figure 4.** Recycling bin (**left**) and locker (**right**).

The main material of the locker is a particleboard with some steel used in the hinges and the handles. The estimated lifetime of the locker is 10 years, and the main reason for replacing it is related to design and fashion considerations. A refurbishment option was offered in the circular offering of the locker, in which the door would be collected after 10 years, repainted, and reinstalled. In the BAU offering, the entire locker would be replaced after 10 years.

The steel for the body of the bin is produced in Sweden, while the medium density fiberboard (MDF) used for the lid is imported from Romania, and the particleboard used for the locker is imported from Latvia. It was assumed that the metal parts would be sent to recycling, and the wood parts would be sent to incineration at the EOL of the product.

The recycling bins are usually sold in pairs; thus, the functional unit used was 2 recycling bins (with a capacity of 85 L each) used for 15 years. The functional unit for the locker was chosen as 1 locker with 9 doors and a dimension of $300 \times 300 \times 47$ cm used for 20 years.

#### 4.2.2. Results

Figure 5a,b shows the overall results of the $LCC_{man}$ and LCA for both the BAU and the circular offerings for the recycling bin and locker, respectively, normalized with respect to the BAU offering. Figure 5c–e shows the contribution of the different life cycle phases to $LCC_{man}$ and GWP. The cost savings were greater for the locker (37%) than for the recycling bin (9%), while the environmental improvement measured as GWP for the circular offering was considerable for both products (47% and 38% for the locker and bin, respectively). Three reasons for the difference in the reduction of the $LCC_{man}$ and LCA can be discerned from these results; namely, (i) distribution costs, (ii) costs and environmental impacts of the dominant phase, and (iii) costs not related to production.

**(i) Distribution costs**: The environmental impacts of distribution were negligible, whereas the distribution cost was a high flat cost, regardless of whether the whole recycling bin was delivered in the BAU offering, or only the lids during repair in the circular offering. This is one reason that there was a difference in the reductions in costs compared to environmental impacts. The influence of distribution costs was especially pronounced for the recycling bin where the distribution cost in the repair phase (for distributing two lids to the customers) became higher than the cost of producing

the product's physical components (Production D and Production R). The GWP of the transportation related to repair activities was negligible.

**(ii) Costs and environmental impacts of the dominant phase**: Similar to Company A, both products received a major contribution to GWP from Production D of durable components in the BAU offering (96% and 52% for the bin and the locker, respectively) and less for Production R (3% and 34% for the bin and the locker, respectively). This explains the large reduction in GWP when the body of the recycling bin, which is made of steel, was reused, despite the production of two extra lids. For the locker, however, the cradle-to-gate GWP of painting the doors, which constitutes a large part of Production R, was around 24% in the BAU offering. This limited the improvement potential of the circular offering, which included repainting the doors. The cost for repainting the doors of the locker was significantly lower than the cost for replacing a whole locker. Thus, the LCC$_{man}$ decreased more when applying the circular offering to the locker than when applying it to the bin.

**(iii) Costs not related to production:** The BAU offering's production-related costs (Production D and Production R) were only 44% for the bin, whereas they were 68% for the locker. This limited the potential for cost reductions for the bin and again demonstrates that the higher the share of production-related costs of the component whose lifetime is prolonged (and, correspondingly, the lower the costs not related to production), the higher the potential for savings, thus making the circular option economically viable.

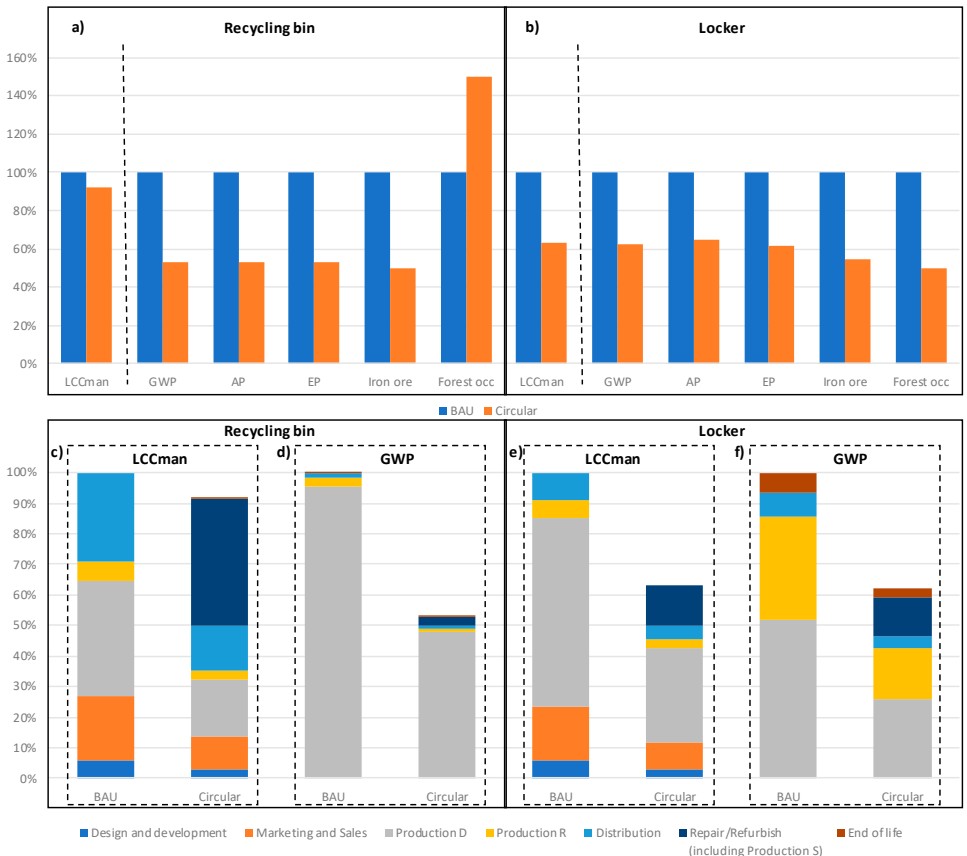

**Figure 5.** LCC$_{man}$ and LCA results for the BAU and circular offerings of the (**a**) recycling bin and (**b**) locker. The contribution of each life cycle phase (in accordance with the categorization in Table 2) to (**c**) LCC$_{man}$ and (**d**) GWP for the recycling bin and (**e**) LCC$_{man}$ and (**f**) GWP for the locker is shown. LCC$_{man}$ is in SEK, GWP is in kg CO$_2$-eq, AP is in kg SO$_2$-eq, EP is in kg PO$_4{}^{3-}$-eq, iron ore is in kg iron ore, and intensive forest occupation is in m$^2$· year. Production D denotes cradle-to-gate production of durable parts, Production R denotes cradle-to-gate production of original wooden components, and repair/refurbish includes the cradle-to-gate production of MDF spare parts and locker paint. All charts are normalized with respect to the BAU offering.

### 4.2.3. Sensitivity Analysis

A different kind of sensitivity analysis was done for the bin and the locker. The number of repairs/refurbishments done during a single lifetime was varied and compared with replacing the product.

Figure 6 shows the results of the sensitivity analysis for the recycling bin and the locker. For both cases, repairing/refurbishing at least twice was considered to achieve an extended lifetime (15 years for the recycling bin and 20 years for the locker) before replacing the product (after 7.5 years for the recycling bin and after 10 years for the locker) becomes more favorable environment- and cost-wise. This means that unless one needs to repair or refurbish more than the breakeven amount during an extended lifetime to keep the quality the same as replacing the product with a new one, then repairing or refurbishing would be the better option. In the case of the recycling bin, there was a huge difference between the GWP and the $LCC_{man}$ (Figure 6a,b, respectively). The environmental analysis shows that up to 35 repair cycles are beneficial in comparison with replacement with a new bin, while costs allow for only two repair cycles before replacement is preferable. This is due to the fact that the environmental impact from the MDF lid is negligible compared to that of the steel body. However, in the $LCC_{man}$, the repair entailed a high transportation cost, which would increase the manufacturer's cost rapidly as more repairs are done. For the locker, the GWP and the $LCC_{man}$ results (Figure 6c,d, respectively) are in line with each other, and the breakeven point occurred at five (5) and four (4) refurbishments, respectively.

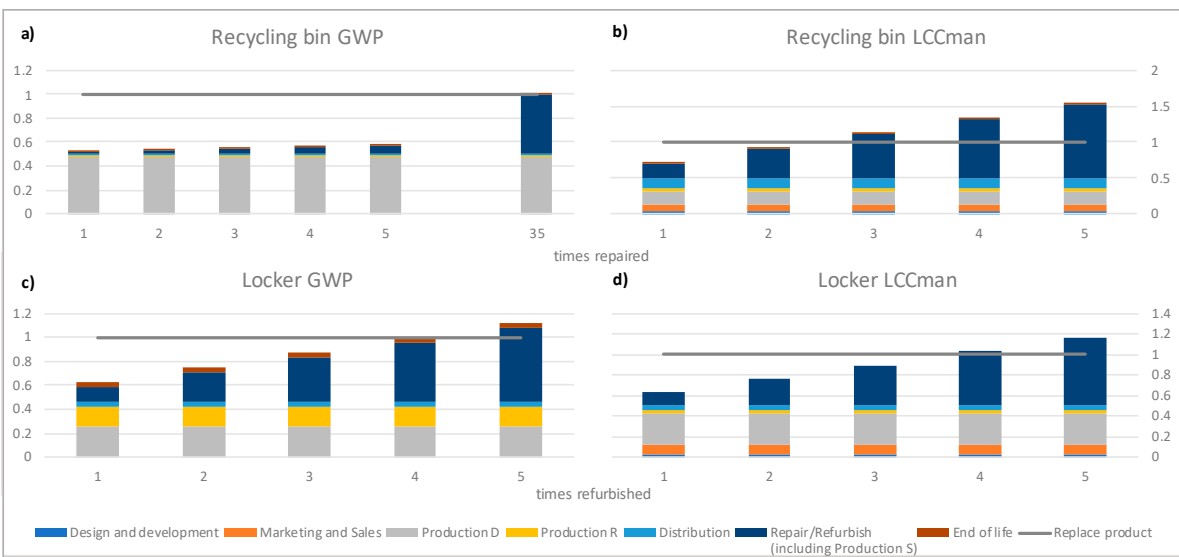

**Figure 6.** The sensitivity analysis results for the varied number of repair/refurbishments (bars) compared to replacing the product with a new one at the EOL (grey line). (**a**) shows the variation of GWP and (**b**) that of $LCC_{man}$ for the recycling bin as the number of repairs is varied. (**c**) shows the variation of GWP and (**d**) that of $LCC_{man}$ for the locker as the number of refurbishments is varied. GWP is in kg $CO_2$-eq, and $LCC_{man}$ is in SEK. All charts are normalized with respect to the product replacement case.

### 4.3. Repair of Waste Inlet

### 4.3.1. Case Product and Company

Company C operates within the waste collection industry, providing automated vacuum waste collection systems for residential areas, hospitals, schools and shopping centers worldwide. It controls and administrates most of the supply chain including design and development, sales, and sourcing (even though partially outsourced, e.g., construction). The system includes an inlet to dispose of the

waste and an underground system with pipes to transport it. The system is sold with an associated service contract.

Only the upper and the lower part of the inlet were studied in the present research (Figure 7, left). The transportation system, including the pipes, fans, and the electricity to drive the fans were excluded from the study. The inlet door (Figure 7, right) breaks down the most often and, therefore, incurs the highest repair costs during the service contract.

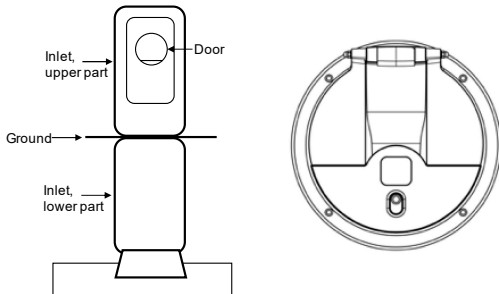

**Figure 7.** Upper and lower part of the inlet (**left**) and inlet door (**right**).

In the circular offering, the door handle was redesigned for ease-of-repair, allowing for replacing the handle alone as opposed to the BAU scenario where the whole door is replaced. In this way, the lifetime for some of the door's components could be prolonged. Both options were done under a maintenance contract with the company, which inspects the inlets at prearranged intervals and guarantees the change of the door or door handle in the event of damage. Both the BAU and circular offerings extend the lifetime of the system through repair but differ with respect to how large a component is replaced at repair. It is estimated that each door must be repaired three times during its lifetime, an estimated 30 years.

The steel for the inlet's body is produced in Sweden. Similar supply chains were assumed for the other components. At the EOL of the product, the metal parts are sent to recycling.

The functional unit is one inlet (upper part and lower part) and one door, used for 30 years.

4.3.2. Results

Figure 8a shows the overall results of the $LCC_{man}$ and LCA for both the BAU and the circular offerings of the waste inlet, normalized with respect to the BAU offering. Starting with the overall system, $LCC_{man}$ decreased by 12%, whereas the GWP decreased by only 1%. This is explained by the higher price for the spare doors (representing around 18% of the total cost), while the environmental impact of producing them is less than 2%. This is the opposite of the other cases where the LCA reductions were more dominant. The two contributing factors, however, are similar, namely, (i) costs not related to production and (ii) costs and environmental impacts of the dominant phase.

**(i) Costs not related to production:** Based on its business model, Company C incurred an additional cost (and environmental burden) that the other companies did not: maintenance cost (Figure 8c,d). Not only did this cost occur regardless of the production, it also occurred whether a door was damaged or not. In addition, the repair cost decreased significantly more than the related environmental impacts (costs were reduced from 18% to 6%, while the GWP was reduced from 2% to 0.5%). This happened mainly because the repair time decreased, and subsequently, lower labor costs were incurred, whereas the environmental savings were mainly due to less Production S.

**(ii) Costs and environmental impacts of the dominant phase:** In contrast to Companies A and B, both the BAU and the circular offerings included a repair scheme. Implementing the circular offering concerned Production S, and Production S made a low contribution to the total GWP of the inlet. The emissions incurred from transportation during the maintenance phase accounted for almost half of the impact in GWP, while most of the other half was derived from the production of steel for the inlet body (Production D in Figure 8c), which was equal in both offerings. Hence, environmental

improvements from changing the door's design (Production R and Production S) were limited and barely seen. Almost the same analogy can be applied to the LCC$_{man}$, the difference being that the cost of producing the door had a larger share in LCC$_{man}$. Alternatively, if the focus is only on the door (production and repair phases), as shown in Figure 8b, an improvement of around 60% can be found for all categories. Note that this is just a zoom-in on the physical parts of Production D, Production R, and the repair of the door and not the full LCC$_{man}$ and LCA. This could be a simplified case where scheduled maintenance trips are removed (e.g., by having an automatic damage sensor), and the product under study is the door.

Finally, similar to Companies A and B, the results of Company C confirm that the higher the share of production-related costs of the component being prolonged in the LCC$_{man}$, the higher the potential for savings and making the circular option economically viable. However, design changes can reduce material cost, but more importantly, labor costs, which were identified as a major cost driver.

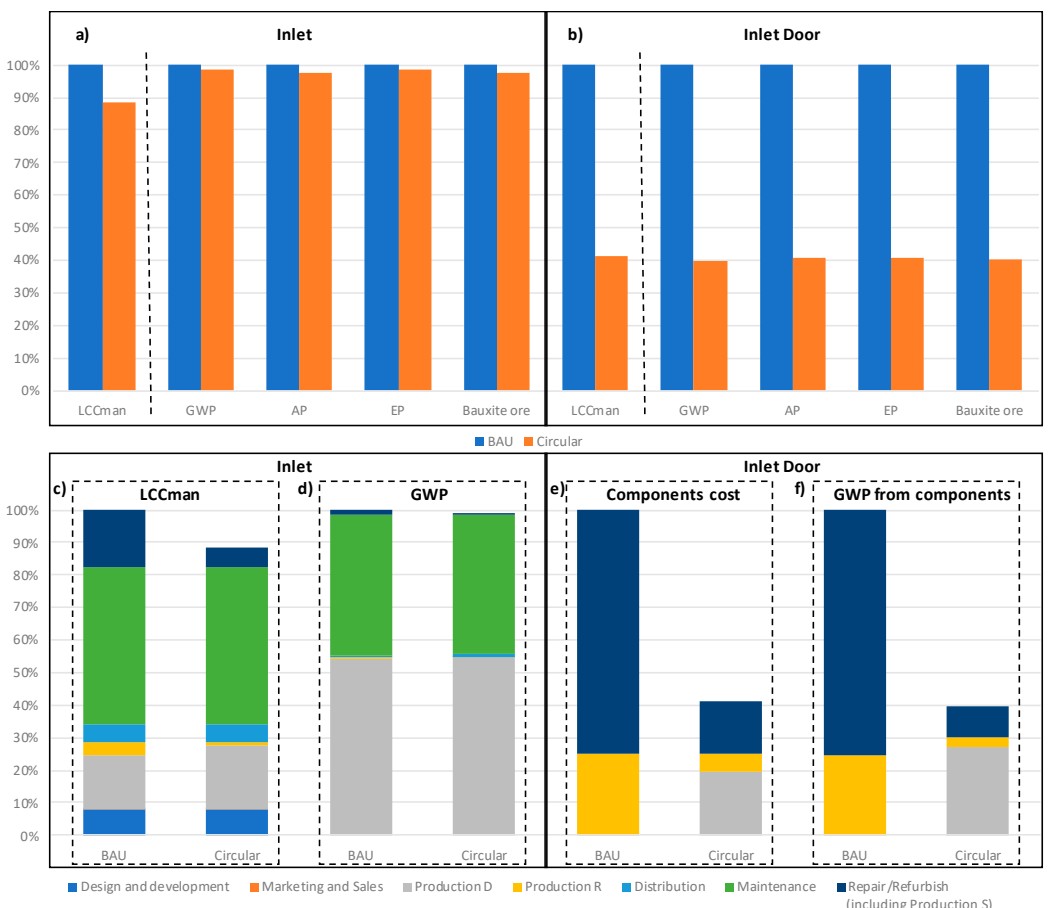

**Figure 8.** LCC$_{man}$ and LCA results for the BAU and circular offerings of the (**a**) inlet and (**b**) inlet door alone. The contribution of each life cycle phase (in accordance with the division done in Table 2) to (**c**) LCC$_{man}$ and (**d**) GWP for the inlet and (**e**) LCC$_{man}$ and (**f**) GWP for the inlet door is shown. LCC$_{man}$ is in SEK, GWP is in kg CO$_2$-eq, AP is in kg SO$_2$-eq, EP is in kg PO$_4^{3-}$-eq, and bauxite ore is in kg bauxite ore. Production D denotes cradle-to-gate production of durable parts, Production R denotes cradle-to-gate production of original door/door handle, while repair/refurbish includes the cradle-to-gate production of the spare door/door handle. All charts are normalized with respect to the BAU offering.

### 4.3.3. Sensitivity Analysis

Both the BAU and the circular offerings for Company C include a repair option, the circular one being more modular. However, it was seen in the results that the new door (circular) contains slightly

more material than the old one (BAU). Thus, it would be worth investigating how many doors should be replaced before the new door becomes more beneficial than the old one. In the original study, it was assumed that the door would be replaced three (3) times during the lifetime of the inlet, and accordingly, this parameter would be varied.

Figure 9 represents the sensitivity analysis for the waste inlet. As depicted, if the door is never damaged during its lifetime, the BAU offering is slightly more environmentally friendly than the circular one (Figure 9a) while having the same cost (Figure 9b). The reason is that the company buys both doors at the same price, even though the more circular door has marginally more aluminum in it. However, from the first door replacement onwards, the circular offering becomes increasingly more beneficial both environmentally and economically.

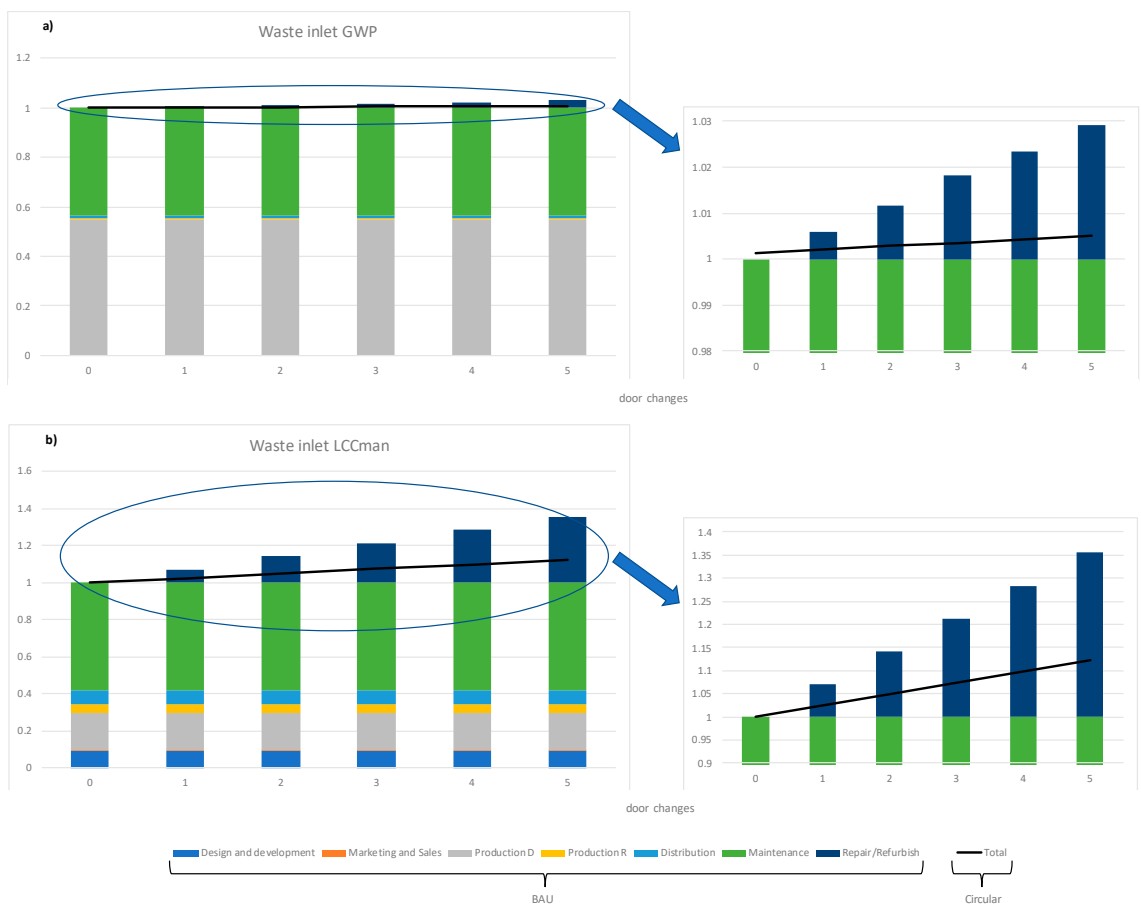

**Figure 9.** The sensitivity analysis results for the varied number of door damages in the waste inlet for the BAU and circular offerings. (**a**) shows the variation of GWP and (**b**) that of $LCC_{man}$ for the waste inlet as the number of door damages is varied. GWP is in kg $CO_2$-eq, and $LCC_{man}$ is in SEK. All charts are normalized with respect to the BAU (no door changed).

## 5. Discussion

### 5.1. Discussion of LCA Results

The five cases in the present study concerned real cases of passive products that were redesigned into more circular offerings so they can be used over a longer period, either as whole products (lockers) or partly (tent, beach flag, recycling bin, and waste inlet). The LCA results documented environmental benefits for all of the more circular offerings and therefore provided quantified cases that support the theoretical motivation of this paper, discussed in Section 2, that prolonging the lifetime of passive durable products has a high potential for environmental benefits.

The proportional environmental improvements were significant for all the products except the inlet and largest for the beach flag and the tent. This is partly because the components from the flag and tent whose lifetime was prolonged contributed the most to the total environmental impacts for the offering (made out of steel, aluminum, and fiberglass) and partly because they were used ten times, reducing the environmental impact of their production by a factor of ten per function. The improvements were smaller for the recycling bin and lockers, because even though their most impacting components' lifetime was extended, i.e., the steel body and wooden parts, respectively, their lifetime was only doubled. This means that they delivered only twice as much function in the circular offering. The inlet had the lowest proportional environmental improvement, because the lifetime of a small component in a large product was prolonged, i.e., parts of the inlets' door. However, when focusing on the component, i.e., the door, the environmental improvement was substantial.

In none of the cases did return transports, repair/refurbish activities, such as repainting the locker doors, or the travels of maintenance personnel outweigh the environmental benefits of the circular offerings. However, according to Kjaer, Pigosso, Niero, Bech, and McAloone [6], achieving resource reduction and avoiding burden shifting between life cycle stages is not enough to decouple economic growth from resource consumption. The third requirement is to mitigate the rebound effect of implementing circular offerings. Such a rebound effect could be repairing/refurbishing more often, assuming that such action is environmentally friendly. However, going back to Figure 6 shows that there is a limit to how many times products can be repaired, and companies should be aware not to exceed the limit. Rebound effects are further discussed in Section 5.3 because many of them are connected to price [57].

The implications for companies are that prolonging the lifetime of passive durable products has a high potential for substituting intensive primary production with lower-impact repair/refurbishment processes, thereby lowering the environmental impacts per unit of function delivered. The results also demonstrated that prolonging the lifetime of components with the highest environmental impacts significantly influences environmental benefits. Finally, the present study showed that increasing the number of uses improves the results, up to a limit, as viewed through the lens of sensitivity analysis.

There are always uncertainties and data gaps in LCA studies. However, since the environmental impact was dominated by the production of the durable parts in this study, the overall finding that extending the lifetime of passive durable products leads to less environmental impact is robust (even if exact numerical results are subject to uncertainty). Nevertheless, there are data gaps in this study for products that contain textile components (the tent and the beach flag). Due to a lack of high-quality data on the impact assessment of textile chemicals, no impact assessment related to toxicity was done, despite the fact that textile chemicals are known as important contributors to the environmental impact of textiles [58]. Consequently, the environmental impact of the textile components is probably underestimated, and hence, the improvement potential of the circular alternative may be overestimated.

*5.2. Discussion of LCC$_{man}$ Results*

The circular economy, as a concept, focuses on decoupling economic growth from resource consumption; this is why circularity cannot be evaluated through the lens of environmental impacts alone. At the firm or product level, i.e., the microlevel, decision makers must be able to see the financial implications of making such a transition. This was emphasized as the main motivator for change by the companies participating in the present study and has been corroborated by several authors, e.g., [25,34,59].

Similar to LCA, extending the lifetime of the five passive, durable products included in this research displayed, without exception, cost reductions for the manufacturer from a life cycle perspective. Other authors have also demonstrated that extending the lifetime of products can reduce life cycle costs, e.g., [24]. Such research remains limited and usually includes active products, and thus, this article contributes to enriching knowledge in this area.

However, it was found that the cost reductions were less pronounced than the reduction of the environmental impact. This result was not unexpected. Although production costs for the components whose lifetime is prolonged are substantial in the BAU scenario, they do not dominate costs to the same extent as they do environmental impact. There are two reasons for this. The first is that an LCC inventory includes other elements, the main one being labor costs. The cost of, e.g., design and development and marketing and sales are mainly labor costs, and therefore, there is no equivalent environmental burden in LCA. Not only can these costs be quite high, but often, they are not reduced in the circular offering because they are not related to production. The addition of services in order to prolong the lifetime of products is expected to incur higher labor costs, e.g., the treatment of the returned product and administration costs. Labor costs are also included in other costs, such as distribution. In contrast to the environmental impacts from transport, which are a direct result of physical aspects, such as the type of vehicle used, distance travelled, and weight of the object transported, distribution costs also include labor costs, such as driver fees and loading fees. This was clearly exemplified in the case of the recycling bins, where a high flat transportation cost was applied, whether delivering a whole recycling bin or only the lids during repair.

The second reason for a difference in the reduction of environmental impacts and costs is that there is not a linear relationship between cost and environmental impact, as discussed in the literature, e.g., [60–62]. For example, production of the event tent components that were reused (aluminum and steel) dominated the environmental impact, but the textile component, which was replaced for every use cycle, was more expensive and, consequently, reduced the potential for cost savings.

Similar to the LCA, the results of the LCC are also subject to uncertainties due to insufficient data, e.g., data on estimations of lifetime and cost. Due to the relatively short lifetime of the products and the quality of the data provided by the companies, these uncertainties were minimized in the present study.

It was found that there is potential to reduce the life cycle costs for the manufacturer and that the higher the share of production-related costs of the component being prolonged in the $LCC_{man}$, the higher the potential for savings and making the circular option economically viable. However, there are challenges due to the effect of other influencing factors, e.g., labor costs. The inlet case, where designing for maintenance significantly reduced LCC compared to just adding maintenance, exemplified and documented the importance of careful planning and designing the offering in order to repeat the full benefits of the PSS [35].

### 5.3. Limitations of LCC and Discussion of Rebound Effects

Although this research found that the manufacturer's cost can be reduced by extending the use of passive durable products, it is important to note that cost alone does not fully portray financial or business implications [61]. Revenues, the dispersion of financial flows over time, and the impact on the financial flows of other actors in the supply chain are also important (ibid.). Revenues, for example, depend on how the circular offerings are priced and, eventually, what customers are willing to pay for them. Customers may perceive circular products as not new and, hence, of less value [57], and therefore not be willing to pay the same price as for a new item. On the other hand, better environmental performance could possibly increase customer willingness to pay, as well as, possibly more importantly to companies, dealings with the same provider instead of investing time and money searching for a new provider will ease the business [35,63]. If willingness to pay changes, apart from costs, revenue should also be compared. Additional aspects come into play for active products, e.g., when a lower cost of operating the product may outweigh a higher price when buying the product. The companies in the present study expected customers to derive equal function from the circular offerings as from the BAU, but customers were not necessarily expected to be willing to pay the same price for repainted locker doors as for a new locker, for example.

Furthermore, if the companies decided to lower the price due to a decrease in $LCC_{man}$, this action might lead to a primary rebound effect [57] where customers consume more of the product.

The possibility of lowering the price and the effect of a lower price on demand, according to the companies' understanding of their market, was discussed. Companies A and B said that they were either going to or were considering reducing the prices of the circular offerings compared to the BAU to stimulate demand. They could not predict whether new customers would be sourced from competitors' market share or whether the size of the market would increase, but they did not expect that their current customer base would increase their demand. Company C said that because repairing an inlet door is not completely covered by the maintenance contract, they thought they might see an increase in demand from price-sensitive existing customers with maintenance contracts as well as customers who are not on maintenance contracts and currently "delay" repairs. Wider rebound effects, such as the effect of a larger supply of offerings on the market and the effect of more available disposable income [57], were not relevant to discuss with the companies and require broader economy-wide assessments.

LCC has an offering perspective; the cumulative effect will also be important for companies implementing changes to several products. These effects have not been presented in this research, which is an important limitation to what can be concluded regarding the economic viability of the circular offering.

Financial barriers, although crucial, are only one set of barriers to a company implementing circular measures or strategies [11,15]. Other barriers, such as a hesitant company culture, might block financially viable solutions from being implemented (ibid.). Since the cases examined in the present study, with the exception of the recycling bin, have either been implemented already or are going to be implemented, they demonstrate that one way to overcome barriers and, consequently, prolong the lifetime of passive durable products is through the addition of services.

Finally, authors have stressed that profitability, or the lack of it, is framed and constrained by the institutional conditions [17] as well as the socioeconomic system in which a CE is embedded [16]. The cases in this research are simple and do not require extensive changes to the companies' supply chains, but many CE measures need to be implemented as part of innovation ecosystems or coordinated action of global supply chains in order to be economically viable.

## 6. Conclusions

This study documented cases where the lifetime of durable passive products has been prolonged through the addition of services and assessed them from an environmental perspective using life cycle assessment (LCA) and from a cost perspective using life cycle costing (LCC). Further, it discussed on a more general level the potential of prolonging the lifetime of passive durable products to contribute to a more circular economy while being economically viable.

The studied circular offerings all had a smaller environmental impact than their business-as-usual counterparts. The reduction in environmental impact was substantial for all cases, excluding the waste inlet (between 45% and 72% in terms of GWP). These findings supported, with real business cases, the hypothesis that the environmental impacts of passive durable products have great potential for reduction by extending the lifetime of the product through the addition of services because environmental impacts are dominated by the material production phase. The impact of return transport and added services was very low and, accordingly, no trade-offs were identified. The main factors that affected improvement potential were, first, the proportion of the impact that is caused by the production of durable components (the larger the proportion, the greater the benefits from lifetime extension), and second, the length of the lifetime extension matters, i.e., the number of times a durable component can be reused or the length of time for which its lifetime is extended, without compromising quality.

The cost to the company responsible for bringing circular offerings to the market was also reduced in all the cases as calculated with LCC from a manufacturer's perspective. However, cost reductions were less pronounced than environmental improvements and ranged from 8% to 37%. This difference is because production does not dominate costs to the same extent it dominates environmental impact. Labor costs accounted for in $LCC_{man}$ have no equivalent in LCA. It was found that some labor costs

were almost fixed in both offerings, e.g., administration and other costs were incurred by the addition of services, thus preventing higher savings from being achieved. The targeted design of a service and an integrated PSS, however, can significantly reduce labor costs, as demonstrated in the waste inlet case.

Therefore, the quantified findings from the five case products demonstrate that prolonging the lifetime of passive durable products through the addition of services can have significant environmental benefits while also being a feasible and cost-reducing circular measure for companies to implement. More case studies and a deeper analysis of rebound effects and the consequences of large-scale adoption should be addressed in future research.

**Supplementary Materials:** The following are available online at http://www.mdpi.com/2071-1050/11/18/4819/s1, Table S1: Overview of materials used and upstream logistics for the beach flag. Table S2: Overview of materials used and upstream logistics for the event tent. Table S3: Overview of materials used and upstream logistics for the recycling bin. Table S4: Overview of materials used and upstream logistics for the storage locker. Table S5: Overview of materials used and upstream logistics for the waste inlet. Table S6: Overview of the manufacturing processes of the five products. Table S7: Overview of the distribution modes and distances for different products. Table S8: Overview of the incineration processes used at product end-of-life. Table S9: Overview of the life cycle impact assessment (LCIA) results for the various products. Table S10: GWP results of the sensitivity analysis for Company A. Table S11: GWP results of the sensitivity analysis for Company B. Table S12: GWP results of the sensitivity analysis for Company C.

**Author Contributions:** Conceptualization, A.-M.T., M.L.K., M.K., and T.S.; methodology, M.K. and M.L.K.; software, M.K. and M.L.K.; validation, M.K. and M.L.K.; formal analysis, M.K. and M.L.K.; investigation, M.K. and M.L.K.; resources, A.-M.T. and T.S.; data curation, M.K. and M.L.K.; writing—original draft preparation, M.K.; writing—review and editing, M.L.K., A.-M.T., M.K., and T.S.; visualization, M.K.; supervision, A.-M.T. and T.S.; project administration, T.S.; funding acquisition, T.S. and A.-M.T.

**Funding:** This research was supported by the Circularis (Circular Economy through Innovation Design) project (No. 2016-03267), funded by VINNOVA, Sweden's Innovation Agency.

**Acknowledgments:** We gratefully acknowledge Aria Soltani, Ban Ahmad, and Ryo Fukushima from Linköping University for their contribution to the LCC calculations. The three companies involved also provided important information and feedback and are gratefully acknowledged.

**Conflicts of Interest:** The authors declare no conflict of interest. The funders had no role in the design of the study; in the collection, analyses, or interpretation of data; in the writing of the manuscript; or in the decision to publish the results.

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
