# Peer review of "Is Prolonging the Lifetime of Passive Durable Products a Low-Hanging Fruit of a Circular Economy? A Multiple Case Study"

_sustainability, doi:10.3390/su11184819_

Round 1

Reviewer 1 Report

This is a well-executed piece of research focused on the environmental and cost aspects of a specific CE strategy (slowing the loops). The general audience would greatly benefit if the innovation ecosystem aspects are added to the framing and discussion, that is, which new connections between actors are needed to prolong the lifetime of the product-cases presented in the papers? How these connections could be established? 

Some references are: 

1. Moreau, V., Sahakian, M., van Griethuysen, P., Vuille, F., 2017. Coming Full Circle: Why Social and Institutional Dimensions Matter for the Circular Economy. J. Ind. Ecol. 21, 497–506. https://doi.org/10.1111/jiec.12598

2. Laurenti, R., Singh, J., Frostell, B., Sinha, R., Binder, C., 2018. The Socio-Economic Embeddedness of the Circular Economy: An Integrative Framework. Sustainability 10, 2129. https://doi.org/10.3390/su10072129

Author Response

Point 1: The general audience would greatly benefit if the innovation ecosystem aspects are added to the framing and discussion, that is, which new connections between actors are needed to prolong the lifetime of the product-cases presented in the papers? How these connections could be established? 

Some references are: 

Moreau, V., Sahakian, M., van Griethuysen, P., Vuille, F., 2017. Coming Full Circle: Why Social and Institutional Dimensions Matter for the Circular Economy. J. Ind. Ecol. 21, 497–506. https://doi.org/10.1111/jiec.12598 Laurenti, R., Singh, J., Frostell, B., Sinha, R., Binder, C., 2018. The Socio-Economic Embeddedness of the Circular Economy: An Integrative Framework. Sustainability 10, 2129. https://doi.org/10.3390/su10072129

Response 1: We agree that the innovation ecosystem aspects could be added to highlight on the connections between different actors. However, in order to narrow the scope of the research, the cases in the article have been chosen because they are simple and do not require the creation of extensive innovation ecosystems. 

Action 1: The choice of cases with limited ecosystem aspects has been clarified on lines 61-62.

The more generic relevance of innovation ecosystem aspects have been touched upon in the background and discussion sections (lines 55-57 and 638-642).

Reviewer 2 Report

- This article is well written and easy to read. I enjoyed reviewing this manuscript. There are major issues that should be corrected.

­-The title reflects the intention and the topic of the article quite well. The abstract is somewhat confusing because it contains some redundancies, and it lacks a clear line of argumentation (problem statement - research question - placement in relevant scientific fields - methodology - key findings - contribution). Therefore, I recommend that the abstract should be rewritten.

-I would also suggest to the authors to extend their literature review regarding the field. It is not clear how this paper contributes to the extant literature. I suggest rewriting the introduction part and critically analyze the literature.For example, in the first paragraph, authors define CE terms, but It does not have any linkage it with the next paragraph start with, “Linear and circular BMs are often”.

-Provide a clear definition of product-service systems (PSSs).. and how does CE initiative support to PSSs. As authors provide a different explanation of PSSs. I suggest authors may explain it in the literature review, but not here in the introduction part. Emphasize what has been done previously in the literature regarding PSS and CE and what actually gaps still need scholarly attention.I also suggest link energy efficient to sustainable development. For example, see Sustainable Development through Energy Management: Issues and Priorities in Energy Savings. Res. J. Appl. Sci. Eng. Technol 7 (2014): 424-429.

-Link the concept of CE and how it supports the Sustainable development.

-Provide appropriate references to the literature in line from 51-58.

-The research question is not clearly articulated to develop one clear research question and objective of the study.Following it, provide one paragraph explaining the methodology, and highlights the expected study contributions.

- I suggest revising the whole introduction part, and provide what previously has been researched in this topic and what is the research gap?

-change this heading to some appropriate, following the journal style. see some sample articles of sustainability journal. change it..2. Motivation

-I’m not exactly certain as to what the best course of action is, and I do not want to impose specific wording on how you structure your propositions. Having said that, I think it is imperative that you do address these concerns regarding the structure of your propositions.

-The methodology is very informative, but it requires major English proofreading. Also, explain the little background of the case selection and company characteristics, assuming the hypothetical name.

-I must commend you for doing a good job with the statistical analysis. However, it is not clear how the case setting and sample has been selected. I am particularly curious to know why the choice to consider such a specific industry of case studies.

-Results analysis is very long, and it seems ambiguous.The author discusses the results in a different style.I suggest discussing under one heading for each case study. For example, 4..2. Results,4.3.2. Results. It can not get reader attention. I suggest authors not to put more focus on writing literature, instead, support your results with the previously established literature and give your arguments. This should be particularly done to ensure the appropriateness and the robustness of both primary sources and secondary resources.

-I also think that the Conclusion section could be strengthened by the provision of more detail about the follow-on research outlined in the new paragraph.

-I suggest authors may rewrite the conclusion part. It must start explaining the purpose and what has been done previously in the domain of studies. Also, provide future research directions precisely in supply chain domain and suggest practical recommendations for the society and academics.

-Before conclusion, provide result discussion in the separate heading. Support the results with findings of previous studies.

Author Response

Point 1: This article is well written and easy to read. I enjoyed reviewing this manuscript. There are major issues that should be corrected.

Point 2: The title reflects the intention and the topic of the article quite well. The abstract is somewhat confusing because it contains some redundancies, and it lacks a clear line of argumentation (problem statement - research question - placement in relevant scientific fields - methodology - key findings - contribution). Therefore, I recommend that the abstract should be rewritten.

Response 2: We agree that the structure of the abstract should follow a consistent structure. We have made extensive changes in order to clearly separate the sections and currently follow the structure in the sustainability template: (1) Background and purpose: L14-17 (2) Methods: L17 -20 (3) Results-Key findings: L21-26 (4) Conclusion: L26-28.

Action 2: Abstract extensively revised

Point 3: I would also suggest to the authors to extend their literature review regarding the field. It is not clear how this paper contributes to the extant literature. I suggest rewriting the introduction part and critically analyze the literature. For example, in the first paragraph, authors define CE terms, but It does not have any linkage it with the next paragraph start with, “Linear and circular BMs are often”.

Response 3: The research issue touches upon a variety of topics

Potential environmental improvements of different types of products/measures (L83-100) PSS and CE connection (L41-50, L107-121) Barriers to CE measures/PSS implementation (L52-62, L117-119) Quantified case studies in PSS and CE from a lifecycle perspective (L122-137)

Key literature from each of these topics is presented in either chapter 1 or chapter 2 or both and a literature gap is presented L122-129. To include all this literature in chapter 1 would in our opinion make it too long as an introductory chapter, hence background and motivation is given in chapter 2. We are also uncertain about which specific topic should have better coverage since the example sentence quoted is not in our text :“Linear and circular BMs are often”.

Action 3: Additional knowledge gaps identified (L122-129)

Point 4a: Provide a clear definition of product-service systems (PSSs).. and how does CE initiative support to PSSs. As authors provide a different explanation of PSSs. I suggest authors may explain it in the literature review, but not here in the introduction part. Emphasize what has been done previously in the literature regarding PSS and CE and what actually gaps still need scholarly attention. 

Response 4a: In L43-44 PSS is defined using a direct quote from a well cited review article: Boehm, M.; Thomas, O. Looking beyond the rim of one's teacup: a multidisciplinary literature review of Product-Service Systems in Information Systems, Business Management, and Engineering & Design. J. Clean. Prod. 2013, 51, 245-260, therefore we do not consider it to be a different definition.

The reason a definition is provided in the beginning is to connect the main concepts CE and PSS that define the research topic. Furthermore it is important to name the three different types of PSS because we return to them later in our proposition in Chapter 1.

The relationship between PSS and CE is described in (L47-59) were we say that CE measures can be operationalized through PSS, but not every PSS is CE. This why we conduct our research because we propose a type of PSS that contributes to CE.

The gap in the literature is further discussed and presented in the chapter 2, Background and Motivation. 

Action 4a: We have reworded some parts in order to clarify the CE - PSS relationship as well as which findings are supported by other authors e.g. L47-50, L53-55.

Chapter 2 has been given a new heading, Background and Motivation in order to clarify that the literature background can be found in this section.

Point 4b: I also suggest link energy efficient to sustainable development. For example, see Sustainable Development through Energy Management: Issues and Priorities in Energy Savings. Res. J. Appl. Sci. Eng. Technol 7 (2014): 424-429.

Response 4b: Energy efficiency is a very important topic in CE and sustainable development but this research focuses on passive products that do not consume energy and materials during the use phase therefore elaborating on energy efficiency is outside the scope.

We realised that some confusion on this issue may be due to the lack of a clear definition for passive products so we have added a definition to the abstract and the text.

Action 4b: Additions in L14 and L64 where passive product is defined

Point 5: Link the concept of CE and how it supports the Sustainable development.

Response 5: There are strong and important connections between CE and SD. These are explored in various academic articles. The relations between these two concepts are however outside the scope of this article, why we In this research and in accordance with the special issue have framed the topic under the CE and PSS concepts and we present results on this relationship. We have gone through the article to make sure that we do not deviate from these concepts. 

Action 5: Checking the whole document and subsequent minor changes

Point 6: Provide appropriate references to the literature in line from 51-58.

Response 6: We have made the referencing of the articles clearer in lines 52 and 55. The findings described are common to all three articles and named as some of the major barriers. These are the articles

Kirchherr, J.; Piscicelli, L.; Bour, R.; Kostense-Smit, E.; Muller, J.; Huibrechtse-Truijens, A.; Hekkert, M. Barriers to the Circular Economy: Evidence From the European Union (EU). Ecol. Econ. 2018, 150, 264-272, doi:https://doi.org/10.1016/j.ecolecon.2018.04.028

Rizos, V.; Behrens, A.; van der Gaast, W.; Hofman, E.; Ioannou, A.; Kafyeke, T.; Flamos, A.; Rinaldi, R.; Papadelis, S.; Hirschnitz-Garbers, M., et al. Implementation of Circular Economy Business Models by Small and Medium-Sized Enterprises (SMEs): Barriers and Enablers. Sustainability 2016, 8, 1212

Mont, O.; Plepys, A.; Whalen, K.; Nußholz, J.L. Business model innovation for a circular economy: drivers and barriers for the Swedish industry–the voice of REES companies. 2017.

Action 6: Rewording and clearer referencing L52-59

Point 7: The research question is not clearly articulated to develop one clear research question and objective of the study. Following it, provide one paragraph explaining the methodology, and highlights the expected study contributions.

Response 7: The research question is to a large extent described by the title and elaborated on in chapter 1.

The methods are is described in section 3, and we feel that including methods when describing the purpose risks blurring the purpose and also to introduce repetition

Action 7: The purpose has been edited, for better clarity (L71-81)

Point 8: I suggest revising the whole introduction part, and provide what previously has been researched in this topic and what is the research gap?

Response 8: The comment is in line with point 3, and we refer to answers to that comment

Action 8: Additional knowledge gaps identified (L122-129)

Point 9: Change this heading to some appropriate, following the journal style. see some sample articles of sustainability journal. change it..2. Motivation

Response 9: The word motivation was chosen because background or literature review does not fully express the intention, which is to pull together different areas of research in order to motivate the study. We propose to change the title to “Background and Motivation”.

Action 9: Changed title of chapter 2 to “Background and Motivation”

Point 10: I’m not exactly certain as to what the best course of action is, and I do not want to impose specific wording on how you structure your propositions. Having said that, I think it is imperative that you do address these concerns regarding the structure of your propositions.

Response 10: We are uncertain about how to interpret this comment.

Action 10: No action was taken.

Point 11a: The methodology is very informative, but it requires major English proofreading. 

Response 11a: The manuscript went through language review before submission.

Action 11a: No action was taken.

Point 11b: Also, explain the little background of the case selection and company characteristics, assuming the hypothetical name.

Response 11b: We agree to the need of an extra background about the companies.

Action 11b: We have added an explanation on the company selection process L142-145 as well as an explanation that the companies are SMEs. Further information regarding the background on the companies has been added to their previous descriptions, in the beginning of the results section L246-248 (Company A), 332-334 (Company B), and 426-428 (Company C).

Point 12: I must commend you for doing a good job with the statistical analysis. However, it is not clear how the case setting and sample has been selected. I am particularly curious to know why the choice to consider such a specific industry of case studies.

Response 12: The research is not based on statistical analysis, but on LCA and LCC of real products deriving from bill of materials provided by the companies. LCC and LCA are by their nature specific to the products/cases. An explanation on the company selection process has been added to L142-145 and the product selection process is in L153-155.

The specific industries that were selected because they sell passive and durable products, which we believe hold a big potential for circular improvements and are usually less focused on in literature.

Action 12: Additions L142-145 and L153-155.

Point 13: Results analysis is very long, and it seems ambiguous. The author discusses the results in a different style. I suggest discussing under one heading for each case study. For example, 4..2. Results,4.3.2. Results. It can not get reader attention. I suggest authors not to put more focus on writing literature, instead, support your results with the previously established literature and give your arguments. This should be particularly done to ensure the appropriateness and the robustness of both primary sources and secondary resources.

Response 13: The results were actually stated under one heading for each case study (4.1.2, 4.2.2, 4.3.2), and the discussion was done the same way to allow for the ease of interpreting what is discussed. We reflect our results to what can be found in literature in Section 5. Discussion

We understand that naming section 4 as “Case study and results” might cause some confusion, so this section was renamed.

Action 13: The naming of Section 4 was changed from 4. Case study and Results to 4. Case studies

Point 14: I also think that the Conclusion section could be strengthened by the provision of more detail about the follow-on research outlined in the new paragraph

Response 14: The last sentence (L673-674) gives an overview of what could be discussed in future research without going deeper into details, because this would be a new research question.

Action 14: No action was taken.

Point 15: I suggest authors may rewrite the conclusion part. It must start explaining the purpose and what has been done previously in the domain of studies. Also, provide future research directions precisely in supply chain domain and suggest practical recommendations for the society and academics.

Response 15: The authors agree that readability of the conclusions is enhanced if the purpose of the study is repeated. However, we believe that the conclusion should focus on the scope of the current study, where the discussion talks about previous studies. The conclusion follows the following structure: (1)Purpose: L644-648 (2)Results: L649-651 (3)Discussion: L651-669 (4)Take-home message to society: L670-673 (5)Future work: L673-674

Action 15: The purpose of the study has been included as an introduction to the Conclusions section (L644-648)

Point 16: Before conclusion, provide result discussion in the separate heading. Support the results with findings of previous studies.

Response 16: In Section 2, results from other studies are presented because they motivate the research. Furthermore, there are not currently many existing studies of passive products. Discussion points (including comparison with findings of previous studies) was performed before the conclusion (sections 5.1, 5.2, 5.3)

Action 16: No action was taken

Round 2

Reviewer 2 Report

Authors have improved substantially, but still it requires minor revision.

-In abstract, in 2nd line, is PSS is circular measure? rewrite if possible.

-In introduction,last paragraph, state one objective of the study, following explain the study contributions in brief. I will suggest authors to follow sustainability guidelines lines.

- Review criticalliterature regarding PSS in introduction

Author Response

Point 1: In abstract, in 2nd line, is PSS is circular measure? rewrite if possible.

Response 1: Extending the lifetime (through repair and refurbish) is the circular measure. PSS is the means to implement that measure. More detailed explanation partly about CE is given in the main text (L37-41).

Action 1: Line 17 has been edited to better reflect the above.

Point 2: In introduction, last paragraph, state one objective of the study, following explain the study contributions in brief. I will suggest authors to follow sustainability guidelines lines.

Response 2: The last paragraph of the introduction (L79-83) has been edited. One purpose is stated, and this has also been given a more “visible” position in the text. The main contributions of the paper are outlined below (bullet points, L84-90)

Action 2: See above

Point 3: Review critical literature regarding PSS in introduction.

Response 3: The comment in the previous review round was “Provide a clear definition of product-service systems (PSSs).. and how does CE initiative support to PSSs. As authors provide a different explanation of PSSs. I suggest authors may explain it in the literature review, but not here in the introduction part. Emphasize what has been done previously in the literature regarding PSS and CE and what actually gaps still need scholarly attention.”

Based on this comment and our answer to it in the previous review round we refer to these key PSS literature in current version

in introduction: (L42-49) Kjaer, L.L.; Pigosso, D.C.; Niero, M.; Bech, N.M.; McAloone, T.C. Product/Service‐Systems for a Circular Economy: The Route to Decoupling Economic Growth from Resource Consumption? J. Ind. Ecol. 2019, 23, 22-35, doi:https://doi.org/10.1111/jiec.12747. Tukker, A. Product services for a resource-efficient and circular economy–a review. J. Clean. Prod. 2015, 97, 76-91, doi:https://doi.org/10.1016/j.jclepro.2013.11.049. Boehm, M.; Thomas, O. Looking beyond the rim of one's teacup: a multidisciplinary literature review of Product-Service Systems in Information Systems, Business Management, and Engineering & Design. J. Clean. Prod. 2013, 51, 245-260, doi:https://doi.org/10.1016/j.jclepro.2013.01.019. Tukker, A. Eight types of product–service system: eight ways to sustainability? Experiences from SusProNet. Bus. strat. env. 2004, 13, 246-260, doi:https://doi.org/10.1002/bse.414.

And these in section 2 (Lines 122-130) Lindahl, M.; Sundin, E.; Sakao, T. Environmental and economic benefits of Integrated Product Service Offerings quantified with real business cases. J. Clean. Prod. 2014, 64, 288-296, doi:https://doi.org/10.1016/j.jclepro.2013.07.047. Sakao, T.; Shimomura, Y. Service Engineering: a novel engineering discipline for producers to increase value combining service and product. J. Clean. Prod. 2007, 15, 590-604, doi:https://doi.org/10.1016/j.jclepro.2006.05.015. Meier, H.; Roy, R.; Seliger, G. Industrial product-service systems—IPS2. CIRP annals 2010, 59, 607-627, doi:https://doi.org/10.1016/j.cirp.2010.05.004. Masood, T.; Roy, R.; Harrison, A.; Xu, Y.; Gregson, S.; Reeve, C. Integrating through-life engineering service knowledge with product design and manufacture. Int. J. Comp. Integr. Manufact. 2015, 28, 59-74, doi:https://doi.org/10.1080/0951192X.2014.900858. Martinez, V.; Bastl, M.; Kingston, J.; Evans, S. Challenges in transforming manufacturing organisations into product-service providers. J. Manufact. Technol. Manag. 2010, 21, 449-469, doi:https://doi.org/10.1108/17410381011046571. Matschewsky, J.; Kambanou, M.L.; Sakao, T. Designing and providing integrated product-service systems–challenges, opportunities and solutions resulting from prescriptive approaches in two industrial companies. Int. J. Prod. Res. 2018, 56, 2150-2168, doi:https://doi.org/10.1080/00207543.2017.1332792. Bigdeli, A.Z.; Baines, T.; Schroeder, A.; Brown, S.; Musson, E.; Guang Shi, V.; Calabrese, A. Measuring servitization progress and outcome: the case of ‘advanced services’. Prod. Plan. Control 2018, 29, 315-332, doi:https://doi.org/10.1080/09537287.2018.1429029.

Therefore, we consider that adequate PSS literature has been covered in both chapters and that we have done so in accordance with your comment in the previous review round. Subsequently we interpret that we should review critical PSS literature that focuses on the specific theme namely the extension of the lifetime of passive durable products.

Action 3: We have added critical PSS literature on extending the lifetime of passive durable products in lines 71 to 78.